

# Ramped thermal analysis for isolating biologically meaningful soil organic matter fractions with distinct residence times

Jonathan Sanderman[1], A. Stuart Grandy[2]

[1] Woods Hole Research Center, Falmouth 02540, USA

[2] Department of Natural Resources and the Environment, University of New Hampshire, Durham 03824, USA

*Correspondence to*: Jonathan Sanderman (jsanderman@whrc.org)

**Abstract.** In this work, we provide a preliminary assessment of whether or not ramped thermal oxidation coupled with determination of the radiocarbon content of the evolved $CO_2$ can be used to isolate biologically meaningful fractions of SOM along with direct information on the turnover rate of each fraction. Using a 30 year time-series of soil samples from a well

characterized agronomic trial, we found that the incorporation of the bomb-spike in atmospheric $^{14}CO_2$ into thermal fractions could be successfully modelled. With increasing activation energy of the fraction, the mean residence time of the fraction increased from 10 to 400 years. Importantly, the first four of five thermal fractions appeared to be a mixture of fast and increasingly slower cycling SOM. To further understand the composition of different thermal fractions, stepped pyrolysis-gas chromatography-mass spectrometry (py-GC/MS) experiments were performed at five temperatures ranging from 330 to

735 °C. The py-GC/MS data showed a reproducible shift in chemistry across the temperature gradient trending from polysaccharides and lipids at low temperature to lignin and microbial-derived compounds at middle temperatures to aromatic and unknown compounds at the highest temperatures. Integrating the $^{14}C$ and Py-GC-MS data suggests the organic compounds, with the exception of aromatic moieties likely derived from wildfire, with centennial residence times are not more complex but may be protected from pyrolysis, and likely also from biological mineralization, by interactions with

mineral surfaces.

## 1 Introduction

Soil organic matter (SOM) consists of a spectrum of material from labile, rapidly cycling compounds to mineral-stabilized molecules that resist degradation for centuries. This spectrum of turnover rates is due to a multifaceted combination of organic chemistries with varying reactivity, various degrees of interaction between organic and mineral phases and greatly

varying microclimates more or less suited to microbial activity (Lehmann and Kleber, 2015; Ruamps et al., 2013; Schmidt et al., 2011). Soil scientists often deal with this complexity by fractionating SOM into what are thought to be more homogenous pools (Christensen, 2001; von Lützow et al., 2007). Similarly, soil carbon cycle models typically divide SOM into conceptual pools with characteristic turnover rates (Manzoni and Porporato, 2009). However, empiricists have been



trying for decades with varying degrees of success to link physically and chemically isolated fractions of SOM to the
conceptual SOM pools in carbon cycle models (Skjemstad et al., 2004; Zimmermann et al., 2007).

Fractionation schemes, employing various physical, chemical, biological or thermal methods, are generally used to reduce
the inherent complexity found in SOM. Both size fractionation and density separation are commonly used to separate
particulate from mineral associated SOM (Elliott and Cambardella, 1991; Golchin et al., 1994; Sollins et al., 2009).
Hydrolysis with strong acid has been shown to isolate SOM that is consistently 100s to 1000s of years older than the bulk
SOM (Paul et al., 2001). Biological fractionation involves the modeling of mass loss or $CO_2$ evolution during a laboratory
incubation experiment (Grandy and Robertson, 2007; Schädel et al., 2013). Baldock et al. (2013), recognizing that fire-
derived pryogenic carbon (PyC) has distinct properties from plant or microbial products, combined physical size
fractionation with the use of solid-state $^{13}C$ NMR spectroscopy to virtually isolate the PyC fraction from each size fraction.
There are advantages and disadvantages to each of these techniques (Poeplau et al., 2018), but even the most detailed
fractionation schemes are unable to isolate homogenous SOM pools (Jastrow et al., 1996; Sanderman et al., 2013; Torn et
al., 2013) which may be as much a result of methodological issues as the fact that there are multiple pathways for SOM
formation (Cotrufo et al., 2013; Sokol et al., 2019).

Thermal analysis techniques, long used in petrochemical exploration and clay mineralogy, offer a promising alternative or
complement to physical- and chemical-based fractionation methods and are increasingly being applied to studies of SOM
stability and loss (Peltre et al., 2013; Plante et al., 2009; Williams et al., 2018). The basic premise of this suite of tools is that
by slowly heating a sample the energy needed to evolve the carbon at different temperatures, whether that comes from
breaking an organic-organic bond or disrupting an organic-mineral bond or other association, can be quantified and that this
energy yield is somewhat related to the energy requirements for enzymatic degradation of SOM (Williams and Plante, 2018).
While research suggests that more complex organic molecules have higher thermal stability (Lopez-Capel et al., 2005; Yang
et al., 2006), contradictory results have also been reported (Rovira et al., 2008). Several studies have also found good
correlations between thermal stability indices and biological stability (Peltre et al., 2013; Soucémarianadin et al., 2018) and
model-derived stable C pools (Cécillon et al., 2018). However, other studies have found that new carbon preferentially
flowed into more thermally stable fractions (Helfrich et al., 2010; Schiedung et al., 2017) suggesting that the relationship
between thermal stability and SOM cycling concepts may not be straightforward.

A recent advance in thermal analysis is the coupling of a temperature-controlled oven to a vacuum line, termed ramped
pyrolysis-oxidation (RPO), enabling collection of evolved gas at distinct temperature regions for subsequent $^{13}C$ and $^{14}C$
analysis (Rosenheim et al., 2008). Unlike interpretation of  other thermal indices, $^{14}C$ is a direct and powerful tracer of soil
carbon cycling by providing information on the age and turnover rate of an isolated fraction of SOM (Trumbore, 2009;
Trumbore et al., 1996). In the first application of the RPO system to SOM, Plante et al. (2013) found that more thermally
stable fractions also contained the oldest most $^{14}C$-depleted carbon.



In this study, we present the first use of ramped thermal analysis with a time series of soil samples to investigate the carbon cycling rate of SOM with increasing thermal stability. The use of a multi-decadal time series of soil samples allows for modeling the uptake of the bomb-spike in atmospheric $^{14}CO_2$ due to nuclear weapons testing into SOM fractions in order to determine the turnover rate of those fractions (Baisden et al., 2013). We further couple the $^{14}$C-based turnover time estimates with parallel chemical characterization of similar thermal fractions using a stepped pyrolysis-gas chromatography/mass spectrometry (py-GC/MS) approach. With the compound specific chemistry data we are aiming to be able to start to explain possible mechanisms controlling the turnover of the thermal fractions thereby providing new insights into the linkages between thermal stability, SOM composition and microbial cycling of SOM.

## 2 Materials and Methods

### 2.1 Trial and soil description

The soil used in this study come from a long-term agricultural research trial evaluating alternative crop rotations located at the Waite Research Institute in South Australia (34.967 S, 138.634 E) with a Mediterranean climate where 80% of the 626 mm of average rainfall occurs during the April-October growing season. The soil is classified as a Rhodoxeralf in the USDA taxonomy (Soil Survey Staff 1999) or a Chromic Luvisol in the WRB taxonomy (IUSS Working Group WRB 2015) with a fine sandy loam texture in the upper horizons, and a mean pH (H2O) of 5.9 and clay content of 18% for the 0-10 cm layer (Grace et al., 1995). Full trial history including management records, agronomic performance and soil data have been reported elsewhere (Grace et al., 1995; Sanderman et al., 2017). These data including monthly climate records can be accessed from the CSIRO Data Access Portal (doi: 10.4225/08/55E5165EC0D29).

For this study, we have chosen to focus on the permanent pasture treatment. This particular trial strip (#29) was under a Wheat/Pea rotation from 1925 until 1950 but then converted to an improved pasture by sowing a mix of annual rye grass, subterranean clover and phalaris in 1950 and was then managed consistently with simulated grazing (i.e. hand mowing) and periodic re-sowing until the end of the trial. In April of 1963, 1973, 1983 and 1993 soil samples from the top 10 cm were collected as a composite of 20 cores taken along the center line of this 90 m long strip trial. Soils were dried at 40 °C for > 48 hr before being stored in glass jars prior to subsampling in 2015. Previous analyses (Sanderman et al., 2016, 2017) suggest that SOC and the proportion in a particulate fraction only varied slightly throughout the 1963-1993 period (Table 1). The large change in $\Delta^{14}$C is due to uptake and loss of the bomb-spike in atmospheric $^{14}CO_2$.

### 2.2 Ramped pyrolysis oxidation

Ramped pyrolysis oxidation (RPO) was performed using the Dirt Burner, a custom built evolved gas system at the National Ocean Sciences Accelerator Mass Spectrometry (NOSAMS) facility, where a sample can be linearly heated under either pyrolyzing or oxidizing conditions. The evolved gases are then oxidized to $CO_2$, measured on an in-line infrared gas analyzer



and trapped for subsequent analysis of $^{13}C$ and $^{14}C$ composition. Initial instrument development is described by Rosenheim et al. (2008) with upgrades described by Plante et al. (2013) (see Hemingway et al. (2017b) for a complete description of current instrument configuration and operating conditions). In this investigation we operated the Dirt Burner only in oxidizing mode. It has been demonstrated that the distribution of activation energies and $^{14}C$ age of thermal fractions for soil are similar under oxidizing and pyrolysis modes (Grant et al. *in review*). The 1973 soil sample was initially run using a fast ramp (20 ° C min$^{-1}$) where only $CO_2$ concentration was recorded. Using an inversion method (Hemingway et al., 2017a), five distinct thermal fractions were identified from this thermogram with temperature ranges of 100-325, 325-400, 400-445, 445-515 and > 515 ° C (Figure A1). Subsequently, 45 mg of each soil sample were combusted on the Dirt Burner using a slow temperature ramp (5 ° C min$^{-1}$) and the evolved $CO_2$ in each of these five fractions was cryogenically collected and purified for subsequent analyses. The RPO fractions were then split for $^{13}C$ analysis on a dual-inlet isotope ratio mass spectrometer (McNichol et al., 1994b) and $^{14}C$ composition via accelerator mass spectrometry after graphitization (McNichol et al., 1994a). Stable isotope data are expressed in $\delta^{13}C$ notation (‰) relative to Vienna Pee Dee Belemnite (VPDB) standard. Radiocarbon data, after using $^{13}C$ data to correct for mass dependent fractionation, are reported from NOSAMS as fraction modern (Fm, where the 1950 atmosphere is assigned a value of 1.0) and subsequently converted to the geochemical $\Delta^{14}C$ notation (Stuiver and Polach, 1977) for bomb-spike modeling.

### 2.3 Pyrolysis gas chromatography-mass spectrometry

We determined the relative percentages and ratios of chemical classes using pyrolysis-mass spectrometry/gas chromatography (py-GC/MS) using methods described previously (Grandy et al., 2009; Kallenbach et al., 2015; Wickings et al., 2011). However, in contrast to previous studies in which we used a single pyrolysis temperature, here we used a 'ramp' or stepped approach by pyrolyzing the same sample at five sequentially increasing temperatures (330, 396, 444, 503 and 735 °C). These temperatures were chosen to correspond with the temperature ranges from the RPO analysis. Briefly, samples were pyrolyzed at each temperature and pyrolysis products transferred to a GC where compounds were separated on a 60 m capillary column with a starting temperature of 40°C followed by a temperature ramp of 5°C min-1 to 270°C followed by a final ramp (30°C min-1) to 300°C. Compounds were transferred from the GC to an ion trap MS where they were ionized, detected via electron multiplier, and identified using a compound library built using the National Institute of Standards and Technology (NIST) database and published literature. Individual compounds were classified by their source as polysaccharide, aromatic, phenolic, protein, and N-bearing (non-proteins), and unknown. The compounds that are in the unknown category are identified but can be derived in nature or due to pyrolysis from different sources (e.g. both protein and aromatic).

### 2.4 Data analysis

The turnover time of each thermal fraction was determined by modeling the incorporation of the bomb-spike in atmospheric $^{14}CO_2$ into each fraction (Baisden et al., 2013). We have applied a steady state soil carbon turnover model to each thermal



fraction where carbon inputs ($C_{in}$) are portioned into each model pool proportional to the fractional distribution of carbon in each pool ($f_{pool}$) with the $^{14}C/^{12}C$ ratio of the previous year's atmospheric $CO_2$ (data from Currie et al., 2011). Carbon losses

follow first-order kinetics with a characteristic decay rate for each pool ($k_{pool}$) and shifts in $^{14}C/^{12}C$ ratio are also affected by radioactive decay ($\lambda = 1.21 \times 10^{-4}$ yr$^{-1}$). Turnover time ($\tau_{pool}$) is simply the inverse of the decay rate for each pool. First, we assumed that each thermal fraction is a single homogenous pool and solve the turnover model to find a single $k$ value by minimizing the sum of square errors (SSE) between observed and predicted $\Delta^{14}C$ data for the four years of data (1963, 1973, 1983 and 1993) for that particular thermal fraction. Initial results suggested that most of the thermal fractions could not be

represented as a single homogenous pool, so we then applied a two-pool model by assuming a fixed $k$ for the fast cycling pool ($k_{fast} = 0.25$ yr$^{-1}$) and allowed the size ($f_{slow}$) and decay rate ($k_{slow}$) of the slow cycling pool to vary while minimizing the SSE between observed and predicted $\Delta^{14}C$ data for each thermal fraction. Model performance was assessed by calculating the root mean square error (RMSE) between observed and predicted $\Delta^{14}C$ values.

The stepped py-GC/MS data are reported as percent relative abundance for each identified compound. We present these data

qualitatively in two ways. First, data from each thermal interval were averaged across years and shifts in the eight major compound classes are shown. Second, all compounds that averaged > 1% abundance across all 20 samples (4 years x 5 thermal intervals) were included in a non-metric multidimensional scaling analysis (e.g. Grandy et al., 2009) after constructing a resemblance matrix using Euclidean distance from the percent abundance data.

## 3 Results

### 3.1 RPO results

All samples produced nearly identical thermograms (Figure 1) suggesting that the activation energies assigned to the 1973 sample (Figure A1) are applicable to samples from all four decades. Fractions 2 and 3 contained the greatest proportion of SOC, with the distribution across all five fractions being consistent between across all years (Table B1). Increasing $\delta^{13}C$ values were found from F1 to F3 with little addition increase from F3 to F5 (Figure 2a). A greater incorporation of the

bomb-spike in $^{14}C$ was found in the lower temperature fractions with no modern carbon seen in the highest temperature fraction (Figure 2b). There wasn't enough sample for $^{13}C$ analysis of F1 or F5 in 1963. Additionally, there was not enough $CO_2$ for $^{14}C$ analysis of F5 in 1963. The $\Delta^{14}C$ value for F1 in 1973 is included in Figure 2b but removed from subsequent bomb-spike modeling.

### 3.2 Bomb-spike turnover modelling results

A single-pool steady-state model to trace the incorporation of the bomb-spike in atmospheric $^{14}CO_2$ into the thermal fractions indicated that turnover times increased from 37 to 386 years with increasing temperature (Table 2). However, with the



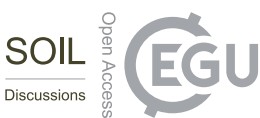

exception of F5 a single-pool model resulted in unacceptably high error with RMSE values ranging from 25 to 45‰ for the other fractions. Visually it was clear that the single pool solution could not capture the dynamics of the increase and subsequent decrease in $\Delta^{14}$C in F1-F4 (Figure C1). A two-pool model with a fast cycling pool ($\tau$ = 4 yr) and a variably sized

slower cycling pool was generally able to capture these dynamics with RMSE values below 13.8‰ (Table 2). With increasing activation energy of the thermal fraction, the proportion of fast cycling carbon decreased (Figure 3a) and the turnover time of the slow cycling fraction increased (Figure 3b).

As an independent first-order check on the reasonableness of MRT results for the thermal fractions, the inventory-weighted MRT of the bulk soil was 16.5 years matching the results of applying a 3-pool turnover model to the bulk $^{14}$C data which

varied from 11-20 years depending on the model structure (Sanderman et al., 2017).

### 3.3 Py-GC/MS results

A total of 172 individual compounds that were identified in one or more samples. Many of these compounds were a minor fraction of only a single sample. When these compounds were classified by source, there were sharp differences in the compound classes dominating each pyrolysis temperature (Figure 4a). Lipids (42.5 ± 2.74 %) and polysaccharides (42.4 ±

2.12 %) dominated the lowest pyrolysis temperature (330 °C). Relative lipid abundance was <10% for all higher temperatures and <1% relative abundance at the highest pyrolysis temperature (735 °C). Polysaccharides (51.2 ± 1.68 %) remained abundant at a pyrolysis temperature of 396 °C and 444 (27.4 %) °C but were <10% relative abundance at the two highest temperatures. At the fourth temperature level (503 °C), N-bearing compounds, proteins, phenols, and unknown compounds dominated the chemical signature and were all ~15-20% relative abundance. Phenols (31.4%) were the most

abundant compound class at the highest temperature (735 °C) while proteins, aromatics and compounds of unknown origin were all >15% relative abundance.

We identified 23 individual compounds that averaged more than one 1% relative abundance averaged across all samples (Figure 4b). These 23 compounds collectively represented from about ~70 to 90% of the total sample relative abundance at the five different temperatures (Table C3). These compounds included three aromatic compounds, one lignin derivative, one

lipid, two nitrogen bearing compounds, two phenols, five polysaccharides, five proteins, and four compounds that fell into the unknown origin. The most abundant compound was hexadecanoic acid methyl ester (palmitic acid) with a mean relative abundance across all samples of about 10% but composing 42% of F1 (Table C1). The next two most abundant compounds were both phenols and included phenol and 4-methyl phenol, both increasing with increasing temperature, followed by the common polysaccharide pyrolysis product furfural, primarily found in the lower temperature fractions.





## 4 Discussion


While this study has only examined one soil under consistent management over four decades, the combined results from the modeling of the incorporation of bomb-spike $^{14}CO_2$ into thermal fractions and the stepped py-GC/MS analysis here suggest that evolved gas analyses can be powerful analytical tools for understanding the complexities of SOM cycling. With increasing activation energy, turnover times increased from decades to centuries (Figure 3) and there was a consistent (i.e.

repeatable across time) strong shift in OM chemistry (Figure 4).

This investigation was partially framed in the context of testing evolved gas analysis as a tool for isolating biologically meaningful fractions of SOM. The results indicate that each of the five thermal fractions contained unique information. Each of the thermal fractions represented pools with a diverse mixture of organic materials with $^{14}C$-based turnover times ranging from 10 to 400 years. A wide distribution of activation energy (Figure A1) was needed to describe each thermal fraction.

Additionally, a one-pool turnover model could not represent the dynamics of $^{14}C$ in all but the most thermally stable fraction (Table 3). This finding of heterogeneity within an isolated fraction also plagues other physical and chemical techniques for isolating carbon fractions (Sanderman et al. 2014; Torn et al. 2013). Here, by combining stable isotope data, $^{14}C$-based turnover modeling and compound specific chemical characterization of the thermal fractions, we have found clues as to reasons for the heterogeneity of SOM within these fractions which provide insights into pathways of SOM formation and

longer-term stabilization.

At the two lowest temperature intervals, where $\delta^{13}C$ values were depleted relative to bulk SOM and $^{14}C$ data suggested rapid turnover rates, the polysaccharides comprised > 40% of the identified compounds. The dominant polysaccharide products at the two lowest temperatures included furfural, 3-furaldehyde, and levoglucosenone. Furfural is a pyrolysis product of hexoses as well as pentoses as well as uronic acids. The hexoses originate from both plant and microbial residues, while the

pentoses are primarily microbial in origin. Levoglucosenone is a pyrolysis product of neutral sugars such as glucose, galactose and mannose (Saiz-Jimenez et al., 1979; Saiz-Jimenez and De Leeuw, 1986). These polysaccharides may be part of partially decomposed residues that dominate the light or particulate fractions of SOM or be part of other soil carbon pools not strongly protected from thermal degradation by association with minerals. Across all thermal fractions, polysaccharide abundance was strongly correlated with the proportion of fast cycling carbon in that fraction.

Identifiable lignin derivatives (e.g. syringol and guaicol) were most abundant in the second and third thermal fractions. While the chemical complexity of lignin likely affords some thermal stability, as suggested by the greater activation energy, these moieties do not appear to be protected from thermal degradation by association with minerals, consistent with other findings that there is little recognizable lignin found in the clay fraction of many soils (Baldock et al., 1997; Grandy et al., 2009; Grandy and Neff, 2008), although amounts may vary across different mineral types and environmental contexts

(Kramer et al., 2012).



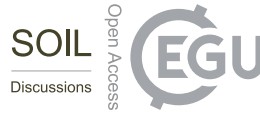

Besides polysaccharides, hexadecanoic acid methyl ester (palmitic acid) was the other dominate constituent of the lowest temperature fraction (Table D1). This compound is a major constituent of microbial cell walls, especially fungi, and has been strongly correlated to both microbial biomass and activity (Zelles et al., 1992). The relative dominance of this lipid at low pyrolysis temperatures suggests that compared to other compounds it is not protected from degradation by either association

with minerals or chemical complexity. There is a growing body of evidence suggesting that a substantial portion of stable SOM in many soils is microbial in origin (Kallenbach et al., 2015; Knicker, 2011; Miltner et al., 2012) which could lead to the misconception that microbial products are inherently more stable than plant-derived compounds. The findings here suggest that the most abundant lipid that we identified, which is most likely of microbial origin, palmitic acid, is not a very thermally stable component in soil with a much faster than average MRT. This may arise because these lipids are not

forming direct covalent bonds with mineral surfaces or forming other complexes (e.g. with metal oxides) that could make them resistant to thermal and biological degradation.

At the third and fourth highest temperatures phenols, proteins and N-bearing compounds became more relatively abundant. These phenols can originate from degraded lignin monomers but also other aromatics of plant or microbial origin. The proteins and other N-bearing compounds are important components of mineral-associated N. The presence of heterocyclic N

(e.g. our furans, pyridines and pyrroles) is a common finding of studies using py-GC/MS that have been identified in prior studies using classical fractionation techniques for 'humic substances' (Schulten and Schnitzer, 1997). The pyrrazoles and pyridines, for example, are believed to largely originate from microbes and Kallenbach et al. (2016) found in an artificial soil system with only glucose inputs that a variety of heterocyclic N compounds can be derived from microbial cells. While secondary 'condensation' reactions that have been proposed to produce humic substances as well as the pyrolysis process

itself have been put forward as potential sources of heterocyclic N these sources are now considered minor contributors compared to microbial synthesis (Paul, 2016).

We have previously found that polysaccharides, proteins and N-bearing compounds are the dominant mineral-associated chemical fractions (Grandy et al., 2007; Grandy and Neff, 2008) as well as an increasing proportion of these compounds with soil depth (Rothstein et al., 2018). Anticipating that these same compounds would also dominate the slowest cycling

highest temperature fraction, we were surprised to find that compounds of aromatic origin or unknown origin, which included compounds such as toluene, the eleventh most abundant compound and that may be derived from pyrolyzing aromatics or proteins, were relatively abundant. There have been reports that pyrolysis of clay fractions can produce matrix effects that inflate the abundance of aromatic compounds (Schlten and Leinweber, 1993). We have seen no detectable evidence for this in our previous studies, which generally show very low abundance of aromatics in fine fractions. The

relatively low clay content and dominance of illite in our soils also point to lower potential for matrix effects. More likely, the high relative abundance of stable aromatics, phenols and compounds of unknown origin can be attributed to the history of fire at this Mediterranean sites. Previous studies at the site indicate that ~30% of the total SOC is made up of pyrogenic

aromatics (Sanderman et al., 2017).

**5 Conclusions**

The findings from this preliminary investigation add to the growing body of literature using evolved gas analysis as a tool for understanding soil organic matter dynamics. The two unique aspects of this work were calculating the MRT of different thermal fractions and relating these MRTs to the chemical composition of each fraction. We found that mean residence time increased with increasing activation energy. Modeling the incorporation of the bomb-spike in $^{14}CO_2$ indicated that the thermal fractions were, except for the most stable fraction, heterogeneous mixtures of fast and slow cycling SOM.

Compound specific analysis demonstrated distinct assemblages of organic compounds were found with increasing thermal stability. These findings together suggest similar thermal activation energies may not equate to similar biological accessibility of the same material and care needs to be taken not to over interpret results from thermal analysis alone. By coupling evolved gas analysis with radiocarbon and compound specific chemical analyses, new insights into the formation, stabilization and fate of SOM may be possible.

**Data availability**

Data from the Waite Permanent Rotation Trial are available for download from the CSIRO Data Access Portal (doi: 10.4225/08/55E5165EC0D29). Radiocarbon and py-GC/MS data are given in appendices B and D.

**Author contribution**

JS conceived the study. JS carried out radiocarbon measurements and interpretation. ASG carried out py-GC/MS
measurements and interpretation. JS and ASG contributed equally to manuscript preparation.

**Competing interests**

The authors declare that they have no conflict of interest.

**Acknowledgments**

We thank NOSAMS director, Mark Kurz, for providing in-kind support for $^{13}C$ and $^{14}C$ analysis of the samples in this study.
We would also like to extend our thanks to Ann McNichol and Mary Lardie for assistance and training in operating the Dirt Burner and Jordon Hemingway for running the activation energy inversion model. Finally, we thank the numerous researchers at the Waite Institute for having the foresight to archive soil samples from the long-term experiment.





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






**Table 1. Previously reported organic matter properties measured on the soils used in this study.**

| Year | SOC (%) | TN (%) | C/N | $\Delta^{14}C$ (‰) | $\delta^{13}C$ (‰) | Size Fractions[1] | | Adelaide Fractions[2] | | |
|------|---------|--------|-----|---------|---------|---------------------|---------------------|-----------|-----------|-----------|
| | | | | | | $f_{>50\mu m}$ | $f_{<50\mu m}$ | $f_{POC}$ | $f_{HOC}$ | $f_{ROC}$ |
| 1963 | 2.80 | 0.25 | 11.3 | 16.7 | -26.5 | 0.27 | 0.73 | 0.17 | 0.54 | 0.29 |
| 1973 | 2.49 | 0.22 | 11.2 | 132.4 | -26.0 | 0.24 | 0.76 | 0.16 | 0.55 | 0.29 |
| 1983 | 2.48 | 0.23 | 10.8 | 118.4 | -26.4 | 0.23 | 0.77 | 0.15 | 0.55 | 0.30 |
| 1993 | 2.46 | 0.24 | 10.4 | 95.6 | -26.6 | 0.23 | 0.77 | 0.16 | 0.55 | 0.28 |

[1]measured distribution of SOC into > and < 50 μm size fractions following dispersion and wet sieving

[2]predicted distribution into particulate, humus and resistant organic carbon (POC, HOC and ROC) fractions (Baldock et al. 2013a) using a mid-infrared spectroscopy based predictive model (Baldock et al. 2013b)






**Table 2. Results from bomb-spike soil carbon turnover modeling.**

| Fraction | E (kJ mol⁻¹) | One-pool | | Two-pool | | | | |
|----------|--------------|----------|------|-----------|-----------|---------|---------|------|
|          |              | $\tau$ (yrs) | RMSE | $f_{fast}$ | $f_{slow}$ | $\tau_{fast}$ | $\tau_{slow}$ | RMSE |
| F1 | 142.0 | 37.3 | 28.6 | 0.38 | 0.62 | 4.0 | 56.16 | 7.1 |
| F2 | 155.2 | 34.1 | 45.2 | 0.30 | 0.70 | 4.0 | 65.80 | 13.8 |
| F3 | 167.5 | 99.9 | 27.1 | 0.16 | 0.84 | 4.0 | 197.2 | 7.9 |
| F4 | 178.1 | 132.4 | 24.9 | 0.16 | 0.84 | 4.0 | 297.3 | 5.3 |
| F5 | 198.0 | 386.3 | 8.9 | 0.00 | 1.00 | 4.0 | 390.4 | 8.7 |






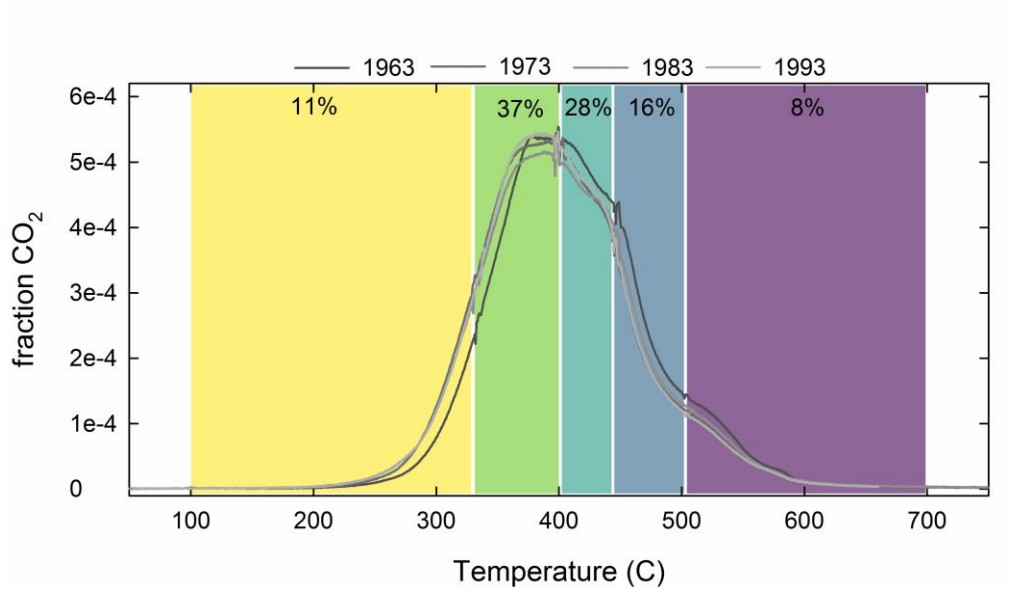

**Figure 1. Proportion of total $CO_2$ evolved with temperature as samples were oxidized with a ramp rate of 5 ° C min$^{-1}$. Thermal fractions are shaded with colors corresponding to Figures 2-4 and mean percent of total is given for each fraction.**




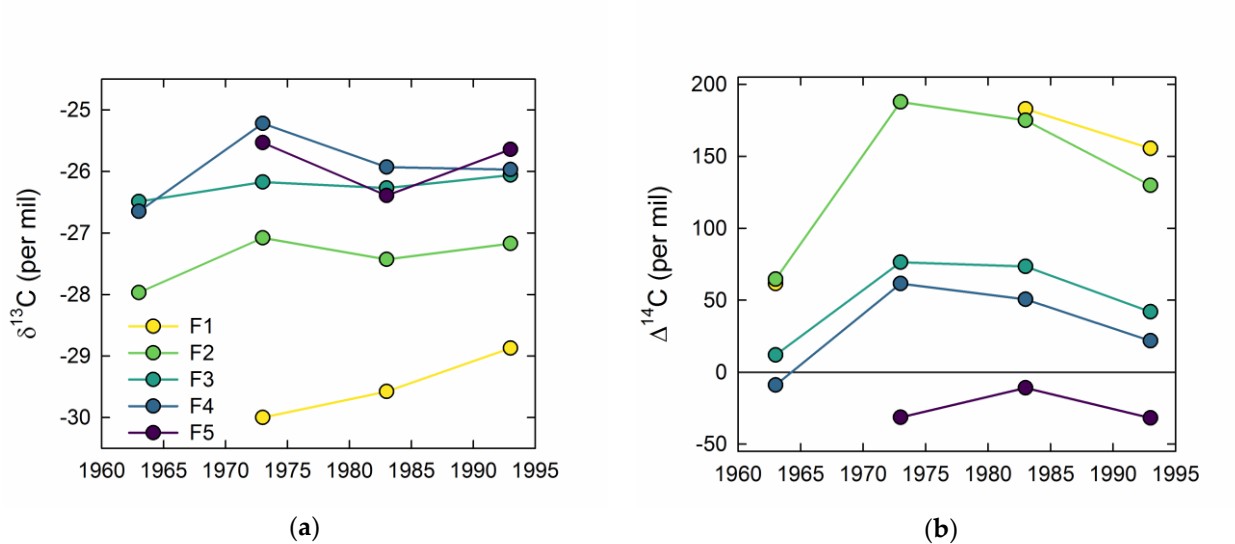

**(a)**          **(b)**

**Figure 2. Variation in (a) $\delta^{13}C$ and (b) $\Delta^{14}C$ in the five thermal fractions across the four decades of soil sampling. $\Delta^{14}C$ value from**
**F1 from 1973 not shown in (b) as this data point was removed as an outlier from the turnover modeling.**





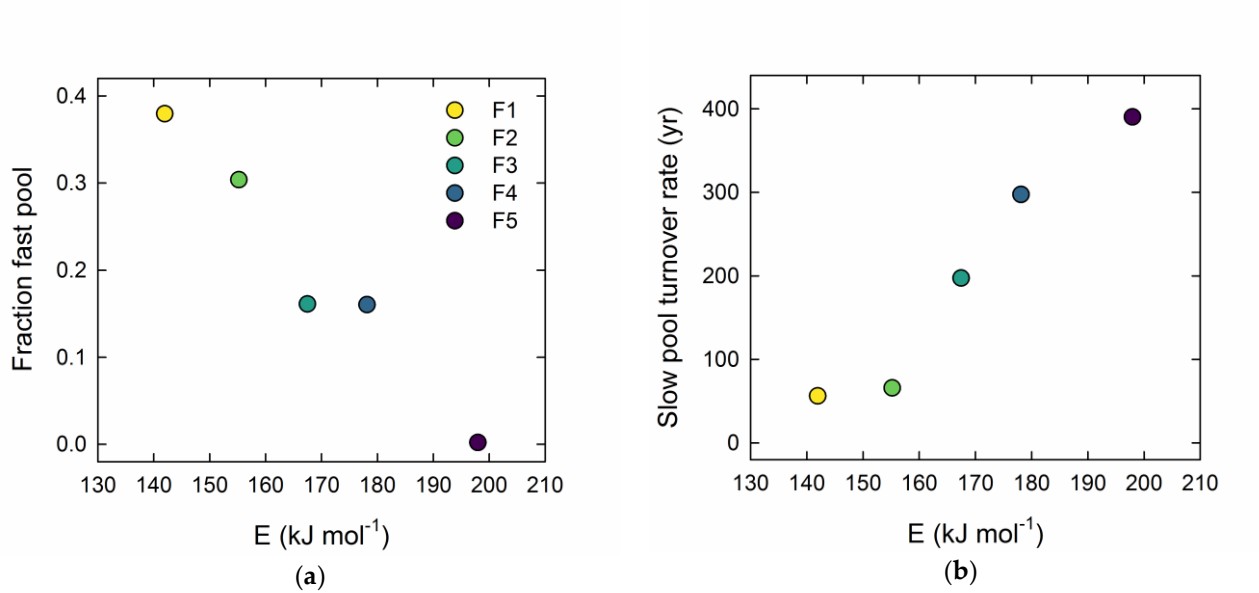

(a)                                                           (b)

**Figure 3. Two-pool $^{14}$C modeling results: (a) proportion of fast cycling carbon in thermal fraction and (b) turnover time (τ) of slow**
**cycling pool plotted as a function of mean activation energy of each thermal fraction.**



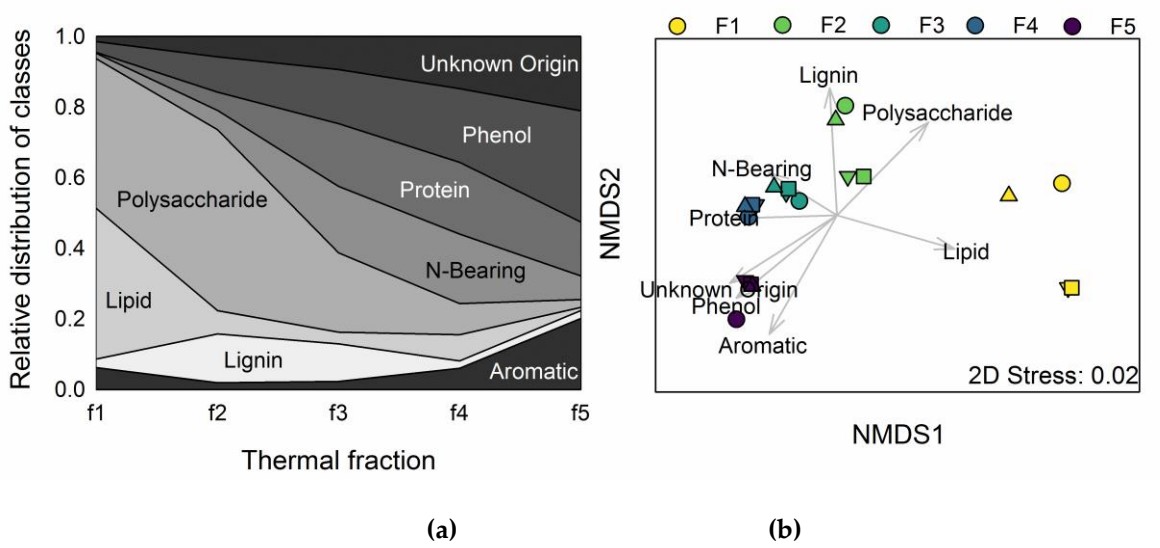

**(a)**      **(b)**

**Figure 4. Stepped pyrolysis gas chromatography-mass spectrometry results: (a) Mean distribution of major compound classes across the four years; (b) Non-metric multidimensional scaling plot of all compounds with > 1% mean abundance (n = 23). In (b) vectors represent correlations with major compound classes and different years are given by different symbols (● 1963, ▼ 1973, ■ 1983, ▲ 1993).**






**Appendix A**

Following acquisition of an initial thermogram (i.e. the trend of $CO_2$ evolution versus increasing temperature) for the 1973 sample, a mathematical inversion method was employed to deconvolve the evolved gas analysis into $N$ number of pools with

distinct activation energy profiles (Hemingway et al. 2017).

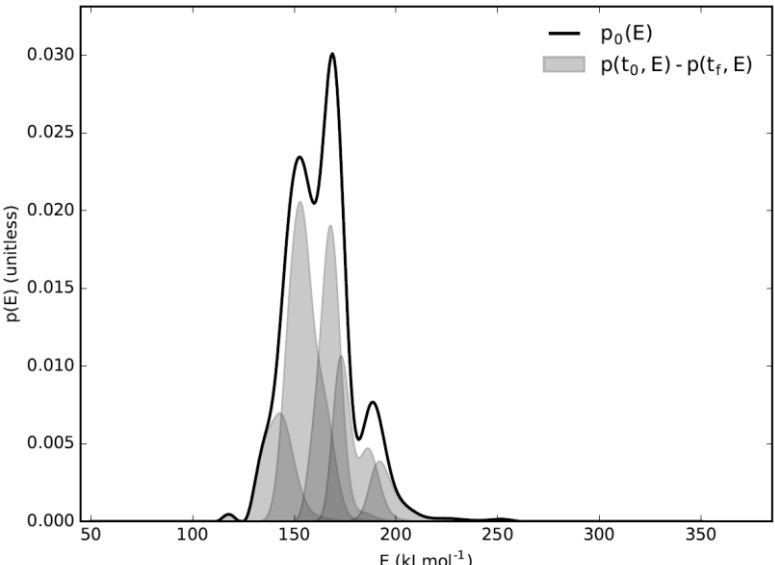

**Figure A1. Proportion of activation energy (E) found with increasing activation energy. Distribution of E within each RPO fraction given by shaded regions.**

**Table A1. Thermal fraction activation energies and distribution (s.d. = standard deviation) based on inversion analysis.**

| Fraction | T0 (° C) | E (kJ/mol) | s.d. |
|----------|----------|------------|-------|
| 1 | 100 | 141.96 | 8.21 |
| 2 | 325 | 155.21 | 7.76 |
| 3 | 400 | 167.47 | 5.98 |
| 4 | 445 | 178.13 | 8.00 |
| 5 | 515 | 197.95 | 13.24 |






**Appendix B**

**Table B1. Ramped oxidation isotope results.**

| Year | Fraction | frac of total C[1] | NOSAMS accession # | F modern | Fm Error | $\delta^{13}C$ (‰) | $\Delta^{14}C$ (‰) |
|---|---|---|---|---|---|---|---|
| 1963 | 1 | 0.08 | OS-131571 | 1.0632 | 0.0023 | n.d. | 61.5 |
| | 2 | 0.34 | OS-131374 | 1.0662 | 0.0021 | -27.97 | 64.5 |
| | 3 | 0.30 | OS-131504 | 1.0135 | 0.0020 | -26.49 | 11.9 |
| | 4 | 0.18 | OS-131505 | 0.9926 | 0.0021 | -26.65 | -9.0 |
| | 5 | 0.10 | no sample | | | | |
| | | | | | | | |
| 1973 | 1 | 0.11 | OS-131506 | 0.9754 | 0.0033 | -30.00 | -27.3 |
| | 2 | 0.41 | OS-131375 | 1.1910 | 0.0024 | -27.08 | 187.7 |
| | 3 | 0.26 | OS-131507 | 1.0793 | 0.0044 | -26.17 | 76.3 |
| | 4 | 0.16 | OS-131508 | 1.0645 | 0.0021 | -25.22 | 61.5 |
| | 5 | 0.07 | OS-131574 | 0.9713 | 0.0026 | -25.53 | -31.4 |
| | | | | | | | |
| 1983 | 1 | 0.11 | OS-131509 | 1.1876 | 0.0026 | -29.58 | 182.9 |
| | 2 | 0.37 | OS-131376 | 1.1796 | 0.0023 | -27.43 | 174.9 |
| | 3 | 0.28 | OS-131377 | 1.0777 | 0.0026 | -26.27 | 73.4 |
| | 4 | 0.17 | OS-131510 | 1.0548 | 0.0022 | -25.93 | 50.6 |
| | 5 | 0.07 | OS-131573 | 0.9930 | 0.0026 | -26.39 | -11.0 |
| | | | | | | | |
| 1993 | 1 | 0.14 | OS-131511 | 1.1615 | 0.0024 | -28.87 | 155.5 |
| | 2 | 0.36 | OS-131378 | 1.1357 | 0.0025 | -27.17 | 129.8 |
| | 3 | 0.28 | OS-131707 | 1.0473 | 0.0027 | -26.06 | 41.9 |
| | 4 | 0.15 | OS-131708 | 1.0271 | 0.0020 | -25.97 | 21.8 |
| | 5 | 0.07 | OS-131783 | 0.9732 | 0.0024 | -25.64 | -31.8 |

[1]proportion of total $pCO_2$ found in each thermal split (i.e. fraction)




# Appendix C

Bomb-spike turnover modeling results for one-pool and two-pool models.



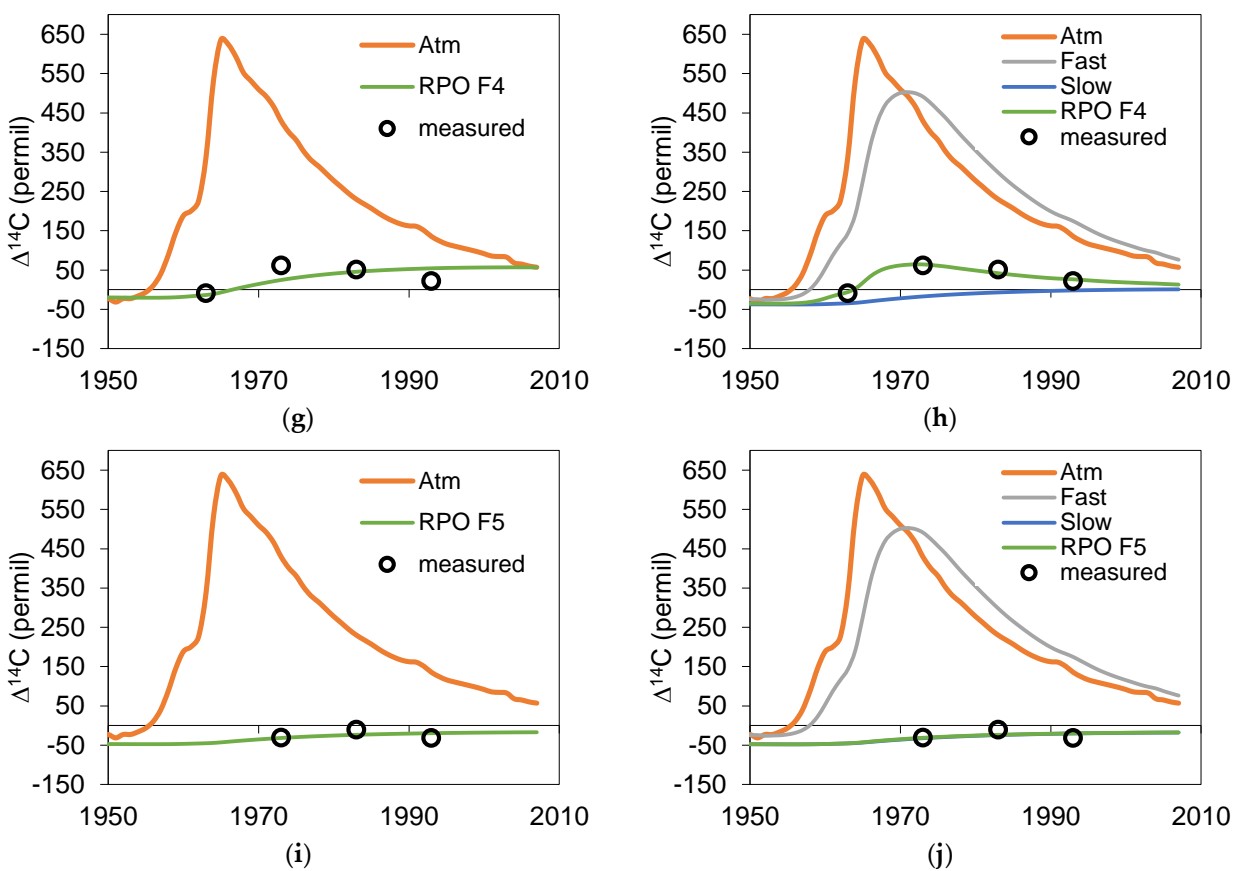

Figure C1. One-pool (a, c, e, g, i) and two-pool (b, d, f, h, j) solutions to a steady-state soil carbon turnover model for the five
thermal fractions. Southern hemisphere atmospheric record given in the background of each panel. For two-pool solutions the
trend in $\Delta^{14}$C are given for the fast and slow pools as well as the overall thermal fraction.



## Appendix D

**Table D1. Heat map of most abundant compound classes (mean across years) identified by stepped Py-GC/MS.**

| Compound | Source | F1 | F2 | F3 | F4 | F5 |
|---|---|---|---|---|---|---|
| Phenol, 3,4-dimethyl- | Aromatic | 0.3 | 0.8 | 1.5 | 2.6 | 2.0 |
| Fluorene | Aromatic | 0.0 | 0.0 | 0.0 | 0.3 | 6.0 |
| Naphthalene | Aromatic | 0.0 | 0.0 | 0.0 | 0.8 | 5.0 |
| Phenol, 2-methoxy- (Guaiacol) | Lignin | 0.9 | 6.1 | 5.4 | 1.0 | 0.0 |
| Hexadecanoic acid, methyl ester (Palmitic acid-C16) | Lipid | 41.5 | 4.9 | 3.1 | 0.4 | 0.0 |
| Pyrazolo[5,1-c][1,2,4]benzotriazin-8-ol | N-Bearing | 0.0 | 1.3 | 11.7 | 13.0 | 1.2 |
| 1H-Pyrrole, 3-methyl- | N-Bearing | 0.0 | 0.9 | 1.9 | 2.5 | 0.6 |
| Phenol | Phenol | 1.0 | 6.9 | 9.1 | 10.5 | 16.7 |
| Phenol, 4-methyl- | Phenol | 2.1 | 3.1 | 6.2 | 10.3 | 14.6 |
| Furfural | Polysaccharide | 16.0 | 13.4 | 4.1 | 0.2 | 0.0 |
| 3-Furaldehyde | Polysaccharide | 10.2 | 8.3 | 0.9 | 0.0 | 0.0 |
| Levoglucosenone | Polysaccharide | 9.0 | 9.2 | 1.1 | 0.0 | 0.0 |
| Benzofuran, 2,3-dihydro- | Polysaccharide | 0.9 | 7.3 | 6.0 | 2.2 | 1.1 |
| Furfural, 5-methyl- | Polysaccharide | 5.7 | 7.8 | 3.1 | 0.5 | 0.0 |
| Indole | Protein | 0.2 | 2.5 | 5.0 | 3.7 | 4.1 |
| Pyrrole | Protein | 0.0 | 1.0 | 3.3 | 3.4 | 0.4 |
| Benzyl nitrile | Protein | 0.0 | 0.5 | 3.0 | 3.4 | 0.3 |
| 3-Methylindole | Protein | 0.0 | 0.4 | 1.5 | 2.3 | 2.8 |
| Styrene | Protein | 0.0 | 0.0 | 0.9 | 3.6 | 2.2 |
| Toluene | Unknown origin | 0.0 | 0.5 | 2.2 | 5.8 | 5.9 |
| 1,3,5-Cycloheptatriene | Unknown origin | 0.4 | 2.8 | 3.4 | 0.7 | 0.0 |
| Phenanthrene | Unknown origin | 0.1 | 0.0 | 0.0 | 0.1 | 6.9 |
| Monobenzone | Unknown origin | 0.0 | 0.9 | 2.6 | 2.2 | 0.0 |
| % of all compound abundance…………………….. | | 88.2 | 78.4 | 76.1 | 69.6 | 69.7 |