# Peer review of "Ramped thermal analysis for isolating biologically meaningful soil organic matter fractions with distinct residence times"

_SOIL, 2019_

## Referee Comment (RC1) · Anonymous Referee #1 · 12 Sep 2019

General comments: soil-2019-44-manuscript-version1-1 The combination of ramped thermal oxidation with 14C determination is a relatively novel approach to study organic matter fractions of different turnover times. Therefore, it is worth to be published in an international journal. Overall, I have problems with the presentation of approach and discussion of data. In my view the authors failed to relate their study to existing knowledge from thermal decomposition studies. This could be overcome by revising the text and involving more literature. Following this it should be stated clearly which results confirm former findings and which really add to the existing knowledge. Specific comments: Page:1 lns 5 ff. I miss state of the art and objective in the beginning of abstract. What does "preliminary" mean in this context? How is "biologically meaningful" defined in this respect? In 10: Statement without prove up to here. What is the "activation energy of the fraction"? Not defined so far ... In 35: py-GC/MS data only can show shifts in the composition of volatilized and ionized fraction - not in the bulk chemistry. Be more precise in descriptions .... In 16: "microbial derived" is not a chemical compound class (e.g., polysaccharides and lipids as well can be derived from microbes, e.g. the latter from microbial cell walls). Ins 17-20: This appears speculative to me (at least not logically following the previous statements). Ins 23-24: What is "multifaceted organic chemistries"? In my view there is only the (one) organic chemistry as a scientific discipline. Page:4 In 110: How was the temperature set up? How was the transfer from Py-oven to GC capillary? Important: Has cooling & condensation been avoided? Describe ionization mode (at least briefly). How did you distinguish between protein-derived and (other) N-bearing compounds? Page:6 In 165: I think this is incorrect; compounds cannot dominate a pyrolysis temperature. You mean the compounds released at a certain temperature. Page:7 Ins 185-190: This in fact is not new: analytical pyrolysis (i.e. the combination of in-source pyrolysis with field ionization mass spectrometry) has shown this many years ago and for numerous samples (published in many articles and reviews). Ins 194-195: This is unproven statement in my view. Which new insight in detail are you referring to? Ins 196-200: All this (esp. thermal lability and possible origin of carbohydrates) in fact is not new but has been published many times years ago. Ins: 205 ff. From these analyses no indication of any association of compounds with minerals can be made. In contrast to author′s discussion higher temperatures for thermal release may origin from binding to minerals. But whether this or chemical complexity of rather intact lignin is the reason for the higher activation energy cannot be judged from the present results. Thus, this discussion should be tuned down or better arguments for the one or the other point of view should be provided. In 209: This has been reviewed already in 2000 (Biol Fertil Soils, 30:399–432). Page:8 This discussion is misleading by restricting to humic substances fractions and the py-GC/MS technique which is not the very best for detecting native N-heterocycles. For discussing this issue first carefully study "Advances in understanding organic nitrogen

chemistry in soils using state-of-the-art analytical techniques". Advances in Agronomy 119 (2013) 83-151 Overall this whole sections is not convincing (only 1 reference for "now considered"). ln 235: "aromatic" is not an origin but a chemical structure property. Page:9 lns 245-254: None of this is a conclusion. It is all summary. Either write true conclusions or omit this whole section.

---

## Referee Comment (RC2) · Anonymous Referee #2 · 2 Oct 2019

General comments: The combination of ramped thermal analysis, radiocarbon analysis, and pyGCMS is an interesting and innovative multi-technique approach to addressing one of the most important, ongoing scientific questions in the SOM community. It is worthy of publication, however there are a few areas for improvement. In particular attention needs to be focused on the correlation between RPO and pyGCMS with regards to reaction artifacts. The discussion is much too brief and would benefit greatly by referencing this work to more of the existing SOM-thermal analysis studies on chemistry, thermal decomposition, mineral-association, etc.

Specific comments: Line 26-28: Specify if this is referring to physical and/or chemical

fractionation. Define or give examples of the homogeneous pools. Define the characteristic turnover rates Line 79: why only the pasture treatment? Line 82: When was the end of the trial? Line 109-114: The methods need to be clarified. It is unclear how the final temperatures were reached, what happened after the ramp to 300C? Were samples held at the final temperature? Also, how would the differences in ramp rates and moisture content between the pyGC (30K/min) and RPO (5K/min) affect the thermal decomposition and consequently the chemical composition of evolved species? Ramp rate has an effect on the formation of combustion/pyrolysis by-products and it should be discussed whether the products evolving at the same temperatures in the two methods are in fact identical. Susott, R., 1980. Effect of heating rate on char yield from forest fuels. Research Note, Intermountain Forest and Range Experiment Station USDA Forest Service INT-295, pp. 1–9. Broido and Nelson, 1975. Char yield on pyrolysis of cellulose. Combustion and Flame, 24 (1975), pp. 263-268. Line 116: these appear to be compound classes, not sources. Line 130: what is the basis for the fast cycling pool rate? Line 143 and 196: The effects of isotopic fractionation during thermal decomposition should be included in the discussion. Or are these values consistent with isotopic differences between compound classes? Benner, R., Fogel, M.L., Sprague, E.K. and Hodson, R.E., 1987. Depletion of 13C in lignin and its implications for stable carbon isotope studies. Nature, 329(6141), p.708. Loader, N.J., Robertson, I. and McCarroll, D., 2003. Comparison of stable carbon isotope ratios in the whole wood, cellulose and lignin of oak tree-rings. Palaeogeography, Palaeoclimatology, Palaeoecology, 196(3-4), pp.395-407. Line 162: It would be interesting to see results for the whole soils. Were there compositional differences between years that might support the changes in MRT? Is there a reason only the mean data is shown? Line 198-201: How does the presence of those pyrolysis products affect the calculated MRT of the higher temperature thermal fractions. Line 206: The activation energy of lignin (or any other compound class) is not shown or discussed. This could easily be added and would offer an interesting comparison between the activation energies of the thermal fractions and the compositional analysis. Williams, E.K., Rosenheim, B.E.,

McNichol, A.P. and Masiello, C.A., 2014. Charring and non-additive chemical reactions during ramped pyrolysis: Applications to the characterization of sedimentary and soil organic material. Organic geochemistry, 77, pp.106-114. Line 232-237: This is very important and needs to be discussed in greater detail. It seems that the combined activation energy of these mineral associated OM and covalent bonds is still smaller than the activation energy of the aromatics measured during thermal analysis and that this is may not be directly reflected in natural/enzymatic systems.

Technical comments: Line 9 and 11: clarify "fraction" to "thermal fraction". Line 38: "virtually" implies nearly/almost or in effect, replace with computationally, statistically, or digitally? Line 60: insert comma between oldest and most Line 71: change to "comes". Also this is a run on sentence and should be split into two. Line 80: why are "Wheat/Pea" capitalized? Line 85: I believe this is the first usage of SOC in the paper; it should be defined here or change to SOM? Replace "a" with "the". Line 146-147: this information should also be in the figure caption. Line 162: omit "that" Lines 174 and 177: there is no "Table C1" or "Table C3", I assume the authors are referring to Table D1. Figure 1: If the y-axis is simply the normalized CO2 signal, relabel as such and delete the tick labels/numbers for clarity. It is very hard to differentiate the greys for the sampling years. Figure 2: Explain all missing data in the caption. It may also be helpful to have the bulk soil data in the figures as well. Figure 4: Capitalize the thermal fractions for consistency. Figure C1: Panel a and b, the 'X' needs to be defined. If representing missing data, why no X in panel i and j? Panel j, are RPO F5 and slow overlapping? Table D1: the shading needs to be defined. Also "source" should be compound class?

---

## Author Comment (AC1) · 9 Dec 2019

Reviewer #1

General comments:

The combination of ramped thermal oxidation with 14C determination is a relatively novel approach to study organic matter fractions of different turnover times. Therefore, it is worth to be published in an international journal. Overall, I have problems with the presentation of approach and discussion of data. In my view the authors failed to relate their study to existing knowledge from thermal decomposition studies. This could be

overcome by revising the text and involving more literature. Following this it should be stated clearly which results confirm former findings and which really add to the existing knowledge.

RESPONSE: We thank the reviewer for constructive feedback and have addressed these overall shortcomings by better integrating our findings into the relevant literature as detailed in our responses to the specific comments below.

Specific comments:

lns 5 ff. I miss state of the art and objective in the beginning of abstract. What does "preliminary" mean in this context? How is "biologically meaningful" defined in this respect?

RESPONSE: We had used the term "preliminary" because this research has focused on only one soil as it evolved over the course of a 30 year field trial and therefore we do not know if the results are applicable to other soil types. We had also used the term because the two pyrolysis techniques are not perfectly aligned resulting in some ambiguity in synthesizing results. We have replaced the word "preliminary" with "first". "Biologically meaningful" is the term that has been applied to the isolation of carbon fractions with distinct turnover times, but we have removed this term in the abstract to avoid having to further define it within the space constraints of an abstract. Otherwise, we feel that the first sentence clearly states the objectives of this paper and we disagree that an abstract needs to include a discussion of the state-of-the-art as that is more appropriate and is included in the introductory material.

ln 10: Statement without prove up to here. What is the "activation energy of the fraction"? Not defined so far ...

RESPONSE: The abstract is a summary of the findings, we disagree that a proof of model fit is necessary to include in the abstract. Upon re-reading this part of the abstract with the reviewer's comments in mind, it is clear that this section can be confusing. We have reworded these two sentences to clarify.

ln 35: py-GC/MS data only can show shifts in the composition of volatilized and ionized fraction - not in the bulk chemistry. Be more precise in descriptions....

RESPONSE: We assume the reviewer is referring to L.15. We clarified with the following edit, 'shift in the chemistry of pyrolysis products'.

ln 16: "microbial derived" is not a chemical compound class (e.g., polysaccharides and lipids as well can be derived from microbes, e.g. the latter from microbial cell walls).

RESPONSE: "microbial derived" is an adjective describing the "compounds at middle temperatures" not the compounds (polysaccharides and lipids) found at low temperatures. We focus a lot of attention in the discussion to the fact that the lipids are likely of microbial origin yet they are the least thermally stable compounds because these compounds are often found to dominate the mineral-associated OM fraction in physical fractionation schemes.

lns 17-20: This appears speculative to me (at least not logically following the previous statements).

RESPONSE: Abstracts often conclude with a broader integrative and forward looking statement that contextualizes the results of this particular study in the broader field (i.e. why should we care about these findings). We feel within the constraint of an abstract this sentence is a good summary how we chose to discuss the findings.

lns 23-24: What is "multifaceted organic chemistries"? In my view there is only the (one) organic chemistry as a scientific discipline.

RESPONSE: This paraphrasing of our original sentence is not accurate but the reviewer's comment points out that this description may not precisely convey the meaning we intended, so we have revised this sentence for clarity. The sentence now reads: "This spectrum of turnover rates is due to a combination of organic matter composition with varying reactivity, various degrees of interaction between organic and mineral

phases and greatly varying microclimates more or less suited to microbial activity".

ln 110: How was the temperature set up? How was the transfer from Py-oven to GC capillary? Important: Has cooling & condensation been avoided? Describe ionization mode (at least briefly). How did you distinguish between protein-derived and (other) N-bearing compounds?

RESPONSE: At each temperature interval (330, 396, 444, 503, and 735), the sample was pyrolyzed at that temperature for 20 s. We have clarified the text this way, 'However, in contrast to our previous studies in which we used a single pyrolysis temperature, here we used a 'ramp' or stepped approach by pyrolyzing the same sample at five distinct, sequentially increasing temperatures (330, 396, 444, 503 and 735 °C; e.g. Hempflig and Schulten, 1990; Buurman et al., 2007; Williams et al., 2014). Thus, the same sample was pyrolyzed five times corresponding with each of these temperatures.

Once the sample was pyrolyzed at that temperature, pyrolysis product was transferred into the GC injector through a heated transfer line (200°C), operated in split flow mode (He flow rate of 1.0 ml min-1, split ratio 50:1, 250°C). This process was repeated for each temperature interval, thus pyrolysis product of one sample was run through the GC/MS 5 times total for each temperature. The pyrolysis product for each temperature interval did not cool because it was injected into the GC through the heated transfer line immediately after pyrolysis. We have now more clearly articulated this in the methods. We have added text to the methods that the separated pyrolysate was ionized in the mass spec at 70 eV in the EI mode with the source temp held at 200°C. We have used the protein class in a number of papers, beginning with Grandy et al. 2007, Soil Biology and Biochemistry. This class consists of previously identified pyrolysis products (e.g. pyridines, pyrroles, indole; Schulten and Schnitzer, 1997, Biology and Fertility of Soils). We have now added the following text to the methods, 'Proteins include pyridines, pyrroles and indole that have been previously identified as pyrolysis products of proteins (Schulten and Schnitzer, 1997; Leinweber et al. 2014).'
ln 165: I think this is incorrect; compounds cannot dominate a pyrolysis temperature. You mean the compounds released at a certain temperature.

RESPONSE: We edited this in line with the reviewer's suggestion.

lns 185-190: This in fact is not new: analytical pyrolysis (i.e. the combination of in-source pyrolysis with field ionization mass spectrometry) has shown this many years ago and for numerous samples (published in many articles and reviews).

RESPONSE: We appreciate the review pointing out that there is existing literature on the topic of soil organic matter chemistry varying with pyrolysis temperature. The exact statement the reviewer is referring to here is a brief summary of the integrated 14C and pyrolysis data which we then spend the rest of the discussion detailing. This is in fact new. However, for the ramped temperature py-gc/ms we have now included the following references in the methods.

Williams, E.K., B.E. Rosenheim, A.P. McNichol, C.A., Masiello. 2014. Charring and non-additive chemical reactions during ramped pyrolysis: Applications to the characterization of sedimentary and soil organic material. Organic Geochemistry 77:106-114.

Hempflig, E. and H.R. Schulten. 1990. Chemical characterization of the organic matter in forest soils by Curie point pyrolysis-GC/MS and pyrolysis-field ionization mass spectrometry. Organic Geochemsitry 15:131-145.

Buurman, P., F. Peterse, G.A. Martin. 2007. Soil organic matter chemistry in allophanic soils: a pyrolysis-GC/MS study of a Costa Rican Andosol catena. European Journal of Soil Science 58.

lns 194-195: This is unproven statement in my view. Which new insight in detail are you referring to?

RESPONSE: If we only wrote this one sentence, we would completely agree with the reviewer's sentiment here but the next three paragraphs expand on this broad statement.

lns 196-200: All this (esp. thermal lability and possible origin of carbohydrates) in fact is not new but has been published many times years ago.

RESPONSE: What is new and novel about this study is the integration of the 14C data with the pyrolysis data. That has not been published on before. We also think the time-series analysis provides an interesting context. We are not making claims that the pyrolysis data is novel in the sense that we are the first users of it or the ramped approach. Just because an approach has been used previously, in this case ramp py-gc/ms or ramp py-oxidation, doesn't mean it should not be used again especially given the fact that soils varies so greatly in composition and reactivity. We agree that there is some existing literature of analytical pyrolysis data on soil organic matter and as detailed in a previous response we have incorporated several of these studies into the discussion section.

lns: 205 ff. From these analyses no indication of any association of compounds with minerals can be made. In contrast to author's discussion higher temperatures for thermal release may origin from binding to minerals. But whether this or chemical complexity of rather intact lignin is the reason for the higher activation energy cannot be judged from the present results. Thus, this discussion should be tuned down or better arguments for the one or the other point of view should be provided.

RESPONSE: This is a fair critique and we have reworded this paragraph to focus more on the fact that lignin does not persist in soils and the 14C data supports this notion.

Ln 209: This has been reviewed already in 2000 (Biol Fertil Soils, 30:399–432).

RESPONSE: We appreciate the reminder to revisit this great paper and have now included the Schulten and Leinweber (2000) paper in our list of references supporting this point.

Page:8 This discussion is misleading by restricting to humic substances fractions and the pyGC/MS technique which is not the very best for detecting native N-heterocycles.

For discussing this issue first carefully study "Advances in understanding organic nitrogen chemistry in soils using state-of-the-art analytical techniques". Advances in Agronomy 119 (2013) 83-151 Overall this whole sections is not convincing (only 1 reference for "now considered").

RESPONSE: Thanks for pointing this out. We've attempted to clarify and expand this part of the discussion, and included several additional references.

ln 235: "aromatic" is not an origin but a chemical structure property.

RESPONSE: Correct, thanks for catching this.

lns 245-254: None of this is a conclusion. It is all summary. Either write true conclusions or omit this whole section

RESPONSE: We have expanded this final paragraph to induce short discussion of important next steps in this line of research but we agree with the reviewer that a standalone conclusions section is not necessary so we have shifted this paragraph back to the Discussion section.

---

## Author Comment (AC2) · 9 Dec 2019

Reviewer #2

General comments: The combination of ramped thermal analysis, radiocarbon analysis, and pyGCMS is an interesting and innovative multi-technique approach to addressing one of the most important, ongoing scientific questions in the SOM community. It is worthy of publication, however there are a few areas for improvement. In particular attention needs to be focused on the correlation between RPO and pyGCMS with regards to reaction artifacts. The discussion is much too brief and would benefit greatly by referencing this work to more of the existing SOM-thermal analysis studies on chemistry,

thermal decomposition, mineral-association, etc.

RESPONSE: We thank reviewer 2 for the constructive feedback. As detailed below we have tried to address the reviewer's concerns about relating the data from the two methods and consistent with reviewer 1's comments, we have expanded the discussion section primarily in relation to setting our work in existing literature. We have purposely chosen not to take a deep dive into range of possible implications for SOM cycling research that our findings may have because this is just an initial investigation on one soil type.

Specific comments:

Line 26-28: Specify if this is referring to physical and/or chemical fractionation. Define or give examples of the homogeneous pools. Define the characteristic turnover rates

RESPONSE: We were referring generically to all fractionation schemes and go on to elaborate in the next paragraph. We have revised this first sentence to define "homogenous pools" to mean "in terms of reactivity, composition or microbial accessibility depending on the particulars of the study". "Characteristic turnover rates" has been reworded as "distinct mean turnover rates".

Line 79: why only the pasture treatment?

RESPONSE: This was a limited investigation and we chose to focus our limited budget on the trial with the greatest carbon flow to maximize our chance of "seeing" the bomb spike in 14C propagate through the thermal fractions. This explanation has been added to this sentence.

Line 82: When was the end of the trial?

RESPONSE: 1996. Sentence revised to include end date.

Line 109-114: The methods need to be clarified. It is unclear how the final temperatures were reached, what happened after the ramp to 300C? Were samples held at the

final temperature? Also, how would the differences in ramp rates and moisture content between the pyGC (30K/min) and RPO (5K/min) affect the thermal decomposition and consequently the chemical composition of evolved species?

RESPONSE: We attempted to clarify this in response to the first reviewer. It seems that the reviewer may be confusing the pyrolysis temperature targets with the GC temperature ramp. The sample was pyrolyzed at one of the set temperature intervals for 20 s. Then, the pyrolysis product was transferred to the GC column where the compounds were separated with a starting temp of 40°C followed by the temperature ramp of 5° C min-1 to 270°C followed by the final ramp (30°C min-1) to 300°C, which was held for an additional 10 min. Hopefully this is now more clear.

Ramp rate has an effect on the formation of combustion/pyrolysis by-products and it should be discussed whether the products evolving at the same temperatures in the two methods are in fact identical. Susott, R., 1980. Effect of heating rate on char yield from forest fuels. Research Note, Intermountain Forest and Range Experiment Station USDA Forest Service INT-295, pp. 1–9. Broido and Nelson, 1975. Char yield on pyrolysis of cellulose. Combustion and Flame, 24 (1975), pp. 263-268.

RESPONSE: While we are not entirely sure what the reviewer is getting at here. We will respond first to the issue that pyrolysis at lower temperatures can produce by-products (primarily char) that are then fully decomposed at a higher temperature. There are multiple lines of evidence that charring is minimal at least in the RPO measurements – first the shape of the thermogram does not suggest a significant pool of very high activation energy OM (i.e. there is no distinct secondary peak as seen in some studies of simple compounds such as the Broido and Nelson 1975 reference the reviewer suggested). Second, the 14C data suggests the highest temperature fraction is the oldest. If there were significant charring of lower temperature OM then the 14C trend would be obscured by inclusion of this charred but young OM. The second part of this comment is that the two methods may not be releasing identical compounds in each temperature interval. This is a potential limitation that we have acknowledged

in the manuscript and one of the reasons that we have considered this a preliminary investigation because this issue is really difficult to both confirm and to rule out entirely.

Line 130: what is the basis for the fast cycling pool rate?

RESPONSE: This was an omission on our part. We have added a description of a simple sensitivity test which was used to determine kfast: "Given the limited number of degrees of freedom, we decided not to allow for the simultaneous optimization of all three parameters. The value for kfast was set at 0.25 yr-1 after performing a simple sensitivity analysis where kfast was varied from 0.1 to 1.0 yr-1 and fslow and kslow were optimized. This exercise suggested that the overall lowest mean RMSE across thermal fractions was achieved with kfast = 0.25 yr-1."

Line 143 and 196: The effects of isotopic fractionation during thermal decomposition should be included in the discussion. Or are these values consistent with isotopic differences between compound classes? Benner, R., Fogel, M.L., Sprague, E.K. and Hodson, R.E., 1987. Depletion of 13C in lignin and its implications for stable carbon isotope studies. Nature, 329(6141), p.708. Loader, N.J., Robertson, I. and McCarroll, D., 2003. Comparison of stable carbon isotope ratios in the whole wood, cellulose and lignin of oak tree-rings. Palaeogeography, Palaeoclimatology, Palaeoecology, 196(3-4), pp.395-407.

RESPONSE: These are a good but very complex points the review raises.

First, isotopic fractionation during thermal decomposition for a complex mixture such as natural organic matter should be minor – the activation energy of 12C versus 13C of any given compounds would result in a few degree shift in the temperature peak for that individual compound. Given the wide thermal range of each of our fractions, this small kinetic effect would not be seen in the 13C data.

Second, it is difficult to compare our results to isotopic differences of different compounds because within each temperature range, there is a mixture of compounds.

SOILD

Interactive
comment

Both of the studies the reviewer cites focus on the strong depletion in 13C of lignin relative to other compounds but lignin is never more than ~10% of any given thermal fraction. Looking through the literature, palmitic acid is often a few per mil lower than the substrate the microbes were feeding on (Abraham et al. 1998) but given we do not know what fraction of the total SOM the microbes were feeding on, we do not want to make this a talking point in this paper. Also it is important to remember that our starting (or bulk) material isn't fresh plant material but rather it is a mixture of aged SOM.

Given that this is such a complex topic since we are not talking about transformations of fresh OM, we prefer not to get into this sort of speculative discussion in this current paper. As such, we have revised line 196 not to mention the 13C data.

Abraham, W. R., Hesse, C., & Pelz, O. (1998). Ratios of carbon isotopes in microbial lipids as an indicator of substrate usage. Appl. Environ. Microbiol., 64(11), 4202-4209.

Line 162: It would be interesting to see results for the whole soils. Were there compositional differences between years that might support the changes in MRT? Is there a reason only the mean data is shown?

RESPONSE: Changes in MRT were across the temperature fractions. The 14C model assumes that MRT doesn't change with time. The compositional differences were so small between years within each thermal fraction that we decided to just present the mean data. The variance about the mean values are given in the text (lines 164-167) and the multivariate plot in Fig 4b shows the individual years.

Line 198-201: How does the presence of those pyrolysis products affect the calculated MRT of the higher temperature thermal fractions?

RESPONSE: We may not be understanding the reviewer's comment but presumably these compounds would have already been released at low temperatures in the RPO procedure (as they are using Py-GC/MS) and would have no effect on the MRT of compounds released at higher temperature.
Line 206: The activation energy of lignin (or any other compound class) is not shown or discussed. This could easily be added and would offer an interesting comparison between the activation energies of the thermal fractions and the compositional analysis.

Williams, E.K., Rosenheim, B.E., McNichol, A.P. and Masiello, C.A., 2014. Charring and non-additive chemical reactions during ramped pyrolysis: Applications to the characterization of sedimentary and soil organic material. Organic geochemistry, 77, pp.106-114.

RESPONSE: The Williams paper presents an excellent example of one of the numerous reasons that comparing thermal profiles from individual compounds to natural organic matter is potentially problematic. Fig 3 in Williams et al. 2014 shows cellulose peak at 360 degrees and a lignin peak at 705 but when an apple leaf that is primarily a mix of cellulose, hemicellulose and lignin is run they get one sharp peak at 440. This finding suggests that the physical structure of a real plant greatly alters the thermogram.

This comment raises a broader point about interpretation of thermal analysis data in soil and sediment systems is complex interplay between inherent activation energy of different compounds and bonding between organic-mineral components as well as organic-organic binding. Interestingly, in the Williams paper there was only a very minor difference in calculated activation energy between cellulose and lignin (88 v. 90 kJ/mol) which contrasts with previous values reported in the literature (taken from Williams Table 2) ranging from 23-361 kJ/mol for lignin and 111-251 kJ/mol for cellulose. Given these large differences across studies, we are hesitant to compare our Ea values which fall squarely in middle of these ranges.

Line 232-237: This is very important and needs to be discussed in greater detail. It seems that the combined activation energy of these mineral associated OM and covalent bonds is still smaller than the activation energy of the aromatics measured during thermal analysis and that this is may not be directly reflected in natural/enzymatic systems.

RESPONSE: This is an excellent point that the reviewer raises – namely, the SOM community generally considers mineral associated OM (MAOM) to have a high barrier to decomposition by extracellular enzymes but the thermal data suggests a lower than average activation energy. This is perhaps a broader critique of thermal analyses measuring something that is not directly relevant to microbial processes. There is a lot of speculation we can do on this point but we feel that is better left until we assemble a more comprehensive dataset. In the concluding paragraph we have added a sentence acknowledging this point and the need for specific follow on research.

Technical comments:

Line 9 and 11: clarify "fraction" to "thermal fraction". RESPONSE: We have now tried to be consistent in the use of thermal fraction throughout the MS.

Line 38: "virtually" implies nearly/almost or in effect, replace with computationally, statistically, or digitally? RESPONSE: "virtually" has been replaced with "mathematically".

Line 60: insert comma between oldest and most Line 71: change to "comes". Also this is a run on sentence and should be split into two. RESPONSE: Done.

Line 80: why are "Wheat/Pea" capitalized? RESPONSE: No longer capitalized.

Line 85: I believe this is the first usage of SOC in the paper; it should be defined here or change to SOM?

Replace "a" with "the". RESPONSE: SOC is only used twice in the entire paper but SOM is used much more often. We have switched to consistently using SOM throughout.

Line 146-147: this information should also be in the figure caption. RESPONSE: Caption has been revised as suggested.

Line 162: omit "that" RESPONSE: Removed. Lines 174 and 177: there is no "Table

C1" or "Table C3", I assume the authors are referring to Table D1. RESPONSE: Thanks for catching this error. Corrected.

Figure 1: If the y-axis is simply the normalized CO2 signal, relabel as such and delete the tick labels/numbers for clarity. It is very hard to differentiate the greys for the sampling years. RESPONSE: Axis and tick marks revised as suggested. We found it difficult to better differentiate the curves but the point we make is that they are essentially identical so perhaps the grey scale doesn't matter.

Figure 2: Explain all missing data in the caption. It may also be helpful to have the bulk soil data in the figures as well. RESPONSE: Caption has been revised. The bulk data is already in Table 1. We decided not to show it hear because there is already a lot of overlap for the 13C data around these mean values.

Figure 4: Capitalize the thermal fractions for consistency. RESPONSE: Revised as suggested.

Figure C1: Panel a and b, the 'X' needs to be defined. If representing missing data, why no X in panel i and j? Panel j, are RPO F5 and slow overlapping? RESPONSE: Caption was revised so that it is clear the x refers to an outlier. Panels (i) and (j) there was no sample to measure so we don't know where to put an x.

Table D1: the shading needs to be defined. Also "source" should be compound class? RESPONSE: The shading needed to be removed as colour is not allowed in tables. Caption has been revised accordingly.

---

## Author Response (AR2)

**Response to editor's comments.**

A very convincing study which was presented to-the-point. Great to see this formal proof that also thermal fractions remain heterogeneous mixtures of fast and slow cycling SOM. The idea to indeed in future also a priory subdivide SOM into mineral and free OM before engaging in a similar RPO/Py-GC/MS analysis seems very appropriate to me. In doing so we might further discover several less composite thermal fractions.

RESPONSE: Thank you for the kind words.

Just two more points where I see room for improvement.

1° As pointed out by referee 1 conditions for pyrolysis were quite different between the 'Py-GC-MS' and off-line pyrolysis for 'Ramped pyrolysis oxidation'. It should be recognized in the discussion that one cannot just assume that the sequential Py-GC-MS was really able to 'mimic' oxidation at various temperatures during ramped pyrolysis oxidation. Of course sources of pyrolysis products ($CO_2$ for RPO and volatilized compounds for Py-GC) will display quite some correspondence but a 'perfect match' seems impossible.

RESPONSE: We have added a short but full paragraph (L254-259) recognizing this limitation and suggest new research to better understand how matched or mismatched the two data sources are.

2° No interpretation is given on the distinct δ13C of the various thermal fractions. Without this it would in fact be best to omit these data. For instance: why are F1 and F2 that similar in their Δ14C but different in their δ13C? Could this be related to the shift in relative proportions of lipids/lignin/phenols?

RESPONSE: Reconciling the 13C and 14C could make for an interesting discussion as there are intriguing trends that might lead to the differential response. In an earlier draft we attempted such a paragraph but in the end felt it was too speculative without further data so decided to remove it. In the end, we have decided that the focus of this paper is on the 14C and py-GC/MS data so we have removed the 13C data from the manuscript. Measurement of 13C is still in the methods section and the 13C data are still reported in the appendix because these data are necessary for correcting the 14C data.

Minor comments:

I can certainly follow that the authors refrained from trying to really explain the time series of OM composition. But perhaps you might still want to comment on the following two points: Fig.1 How could you match the lower thermal stability of the >1963 soils with their lower OC%? Seems contradictory. + in in 1973: Δ14C was -27.3 ‰ –> reason?

RESPONSE: Good points for us to clarify in the paper.

Without replication in the RPO measurements it is difficult to know if the shift in the 1963 thermogram compared to the 1973-1993 thermograms is significant or not. We could not explain the low 14C value for the fraction of the 1973 sample and treated it as an outlier in the turnover modeling. Unfortunately, we did not have the financial resources to rerun this sample.

L117 superscript for -1

RESPONSE: Corrected.

L217 studies'

RESPONSE: Corrected.

L220 dominant?

RESPONSE: Yes, dominant not 'dominate'

L238 something wrong with this sentence here

RESPONSE: Yes, this sentence as written did not make sense. We have revised to: "Secondary chemical reactions in soil (de Assis et al., 2012) and wildfire (González-Pérez et al., 2004) as well as the pyrolysis process itself (Hatcher et al., 2001) are potential sources of heterocyclic N, but direct plant and microbial inputs also make a key contribution to this pool (Leinweber et al. 2014; Paul, 2016)."

352 'soil comes from'

RESPONSE: We're not sure what line is being referenced here. L352 is in the references.

L252 'site'

RESPONSE: Corrected.

Fig. 1 Difficult to discern thermograms for the different years: use colour perhaps instead of shades of grey. °C instead of C

RESPONSE: Because color is already used in the background to differentiate the 5 fractions, we cannot use color for the plots but we have revised the figure to use different dot-dash patterns for the 4 thermograms. 1973-1993 are nearly identical so regardless of color or line pattern these would be hard to differentiate, but now 1963 plot should be much easier to pick out from the others.

Table 2 -> units for τfast and τslow + perhaps in the caption repeat meaning of τ

RESPONSE: Units were added. τ was defined below the table

Table b2 two different notations are used for the fraction of modern C: 'F modern' and 'Fm' —> keep one.

RESPONSE: Corrected to F modern in both column headers.